# Inducing Human-like Biases in Moral Reasoning Language Models

**Austin Meek** *
University of Delaware

**Artem Karpov** *
Independent

**Seong Hah Cho** *
Independent

**Raymond Koopmanschap**
Independent

**Lucy Farnik**
University of Bristol

**Bogdan-Ionut Cirstea** †
Independent

## Abstract

In this work, we study the alignment (BrainScore) of large language models (LLMs) fine-tuned for moral reasoning on behavioral data and/or brain data of humans performing the same task. We also explore if fine-tuning several LLMs on the fMRI data of humans performing moral reasoning can improve the BrainScore. We fine-tune several LLMs (BERT, RoBERTa, DeBERTa) on moral reasoning behavioral data from the ETHICS benchmark Hendrycks et al. [2020], on the moral reasoning fMRI data from Koster-Hale et al. [2013], or on both. We study both the accuracy on the ETHICS benchmark and the BrainScores between model activations and fMRI data. While larger models generally performed better on both metrics, BrainScores did not significantly improve after fine-tuning.

## 1 Introduction

Recently, multiple papers have shown surprising similarities between the internal representations in biological brains and those in artificial neural networks, in multiple domains and for multiple tasks; see e.g. Huh et al. [2024] for a review and for some potential theoretical explanations of why one might expect this, including with increasingly powerful machine learning models. However, to the best of our knowledge, no previous work has analyzed potential analogous representational similarities in the domain of moral reasoning, nor whether the degree of similarity might be increased by using human neural data (e.g. fMRI).

In this work, we undertake the first such measurement (BrainScore) of the similarity of the internal representations of biological brains (measured through fMRI) and large language models (LLMs), in the domain of moral reasoning (in a task which is also partially relevant to Theory of Mind). We also study whether fine-tuning the LLMs on a train set of the corresponding neural data helps with improving the BrainScore on a separate test set. While our attempts here proved unsuccessful, we think this is an important problem to study and that increasing said alignment might be important for the AI alignment problem Christian [2020].

We next discuss related works in section 2, our methodology in section 3, the results we obtained in section 4, and finally in section 5 conclude and discuss some potential future work.

---

*Equal Contribution
†Corresponding author: cirstea.bogdanionut@gmail.com

Preprint.

## 2 Related Works

There is a growing interest in brain-model alignment work, here we only provide a brief overview. For a more detailed survey, see Sucholutsky et al. [2023], especially section 4.3.2, and Schrimpf et al. [2020] for a systematic approach to collecting and scoring many models. Earlier work on fine-tuning transformers to predict fMRI data has found that adding MEG data also helps Schwartz et al. [2019]. Other work Aw and Toneva [2023] focused on fine-tuning models on the much larger Booksum dataset Kryściński et al. [2022], which they found increased alignment. Dapello et al. [2022] used rhesus macaque neural data and showed improved alignment with human neural data and greater adversarial robustness.

To the best of our knowledge, we are the first to attempt increasing brain-model alignment on moral reasoning neuroimaging data.

## 3 Methodology

### 3.1 Benchmark and Dataset

To quantitatively measure the moral reasoning performance of the fine-tuned LLMs, we used the common sense category of the ETHICS benchmark Hendrycks et al. [2020], which consists of multiple choice questions rather than free form responses. To predict an answer, we use a linear transformation layer (a CLS head or just a head) attached to predictions (logits) for a classification token, "[CLS]", of a base model.

We used the fMRI dataset, 'Moral judgments of intentional and accidental moral violations across Harm and Purity domains', from Koster-Hale et al. [2013]. Human subjects were given a series of scenarios describing moral, immoral, and neutral actions across a wide variety of scenarios, and then answered on a 1-4 scale how moral or immoral each action was. Koster-Hale et al. [2013] was approved by an IRB and subjects were paid and gave written, informed consent.

### 3.2 Data Pre-processing and Analysis

We used a pre-processed version of the dataset from Thomas et al. [2023], specifically the version fit with the DiFuMo atlas Dadi et al. [2020] with a dimensionality of 1,024 regions of interest (the maximum number of dimensions DiFuMo provides).

Because of the high granularity of our chosen atlas, we used NeuroSynth Yarkoni et al. [2011], a tool for meta-analysis conducted over thousands of fMRI studies to isolate regions consistently activated during experiments, to map activations to specific themes. We conducted our analyses on four regions related to Theory of Mind, moral reasoning, language, and vision. We used vision as the control group as we expected scores there not to increase (see Appendix A). We visualized the relationship between the fMRI and LLM activations on the cortical surface (see Appendix A) using the Coefficient of Determination (CoD). The CoD scores were then negative log transformed and the weighted average of the parcel scores were plotted at each vertex, since the DiFuMo atlas is probabilistic with overlapping boundaries.

### 3.3 Models and Fine-tuning Procedure

We focused on encoder models (BERT-based) due to computational constraints and since this was a classification task. Additionally, encoder models originally showed better results on the ETHICS dataset Hendrycks et al. [2020]. Overall we used four models, BERT-base-cased and BERT-large cased (108 and 333 million parameters) Devlin et al. [2019], RoBERTa-large (355 million parameters) Liu et al. [2019], and DeBERTa-v2-xlarge (864 million parameters) He et al. [2021].

For fine-tuning, we used the HuggingFace library Wolf et al. [2019] to train additional heads of dimensionality 1,024 (to match the DiFuMo atlas) on top of the classification token, "[CLS]". We also train heads to predict the ETHICS benchmark Hendrycks et al. [2020]. We report fine-tuning on ETHICS only and with the addition of fMRI data in Table 1. In total, we ran 450 fine-tuning runs, totaling 292 hours of training for 1,082 different models (not all shown in the results section).

| Model | On Ethics only | Runs | CS Hard Set, % (95 CI) | | CS Test Set, % (95 CI) | |
|---|---|---|---|---|---|---|
| | | count | mean | max | mean | max |
| BERT-base | | 35 | 47.3 (0.0, 53.9) | 55.5 | 57.0 (48.8, 70.5) | 73.7 |
| BERT-base | Yes | 7 | 52.3 (50.0, 55.3) | 55.4 | 58.3 (50.0, 71.0) | 71.7 |
| BERT-large | | 28 | 53.6 (49.4, 59.0) | **61.8** | 62.0 (48.5, 78.7) | 85.4 |
| BERT-large | Yes | 16 | 52.5 (48.2, 58.8) | 58.8 | 59.2 (43.9, 78.8) | 79.3 |
| RoBERTa-large | | 4 | 66.1 (51.5, 72.4) | **72.5** | 80.6 (53.0, 91.4) | **91.4** |
| RoBERTa-large | Yes | 18 | 65.7 (49.8, 73.8) | **74.1** | 79.0 (49.7, 91.6) | **91.8** |
| DeBERTa-v2-xlarge | | 9 | 51.8 (45.9, 67.4) | 70.8 | 52.9 (49.9, 66.6) | 70.0 |
| DeBERTa-v2-xlarge | Yes | 3 | 59.5 (49.8, 77.4) | 78.8 | 64.1 (49.9, 90.2) | 92.3 |

Table 1: Results of fine-tuning four different models on the Commonsense split of the ETHICS Hendrycks et al. [2020] dataset. Bolded values are those higher than reported by the original authors. Values are for models fine-tuned on ETHICS only if stated or otherwise fine-tuned on both ETHICS and fMRI data. Parentheses indicate a two standard deviation confidence interval.

## 3.4 Brain Scores

Our 'brain-score' metric is based on similar metrics found within the broader NeuroAI literature, such as in Schrimpf et al. [2020], Aw and Toneva [2023], *inter alia*. We use the Pearson's correlation coefficient (PCC) to measure the correlation between predicted brain activity and actual brain activity. For some moral scenario given to a subject, we sample the fMRI data at several time points, taking the hemodynamic lag into account. This data has been fit to the DiFuMo atlas, resulting in 1,024 ROIs at the time points sampled. We do this over a collection of similar examples, and then fit a regression model to predict this brain activity. The PCC between the predicted response and the actual held out brain activity gives us the brain-score metric. We can also do this on a layer-by-layer basis and aggregate over all layers to provide a single brain-score for the whole model, which we provide in Table 2.

## 4 Results

Our results indicate that improving brain-model alignment on moral reasoning by fine-tuning on relevant fMRI data does not consistently improve accuracy on ETHICS Hendrycks et al. [2020]. While our fine-tuning procedures do improve accuracy on the Commonsense split of ETHICS (bolded values in Table 1 are higher than those reported by Hendrycks et al. [2020]), we could not improve accuracy by fine-tuning on the fMRI data only or on a combination of fMRI and ETHICS, compared to fine-tuning purely on ETHICS.

To thoroughly test these results, we also used a variety of sampling methods to pull from the fMRI data, as shown in Table 3 and Figure 2 in Appendix B. AVG indicates an average of all time points, LAST indicates the time point at the hemodynamic lag before the last time point, MIDDLE indicates the middle point, and SENTENCES indicates four points in a scenario, which match the end of the four sentences read by the subject. We find that LAST tends to produce the best accuracy.

We generally find that larger models are more performant overall, a finding also reported by Hendrycks et al. [2020]. This additionally holds with the brain-score metric, which measures brain-model alignment. However, we were also unable to significantly improve our brain-score metric beyond the pre-trained models, as shown in Table 2. This finding is also consistent in layer-wise scores across each of the three models; see Appendix A for further brain-score details (including region specific information, such as Theory of Mind ROIs) and cortical mappings.

## 5 Conclusion and Future Work

While we were unable to significantly increase brain alignment on moral reasoning through fine-tuning methods, we do believe that our results can be of use for downstream work. Firstly, we believe that our work is ample evidence for the importance of gathering more data on moral reasoning and for more niche tasks in general if future researchers want to increase brain-model alignment

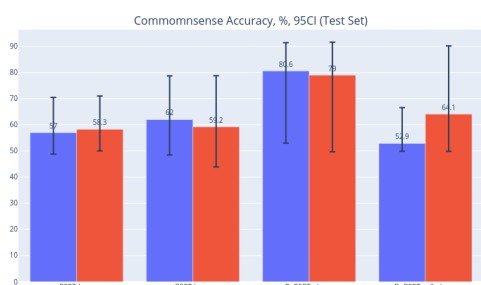
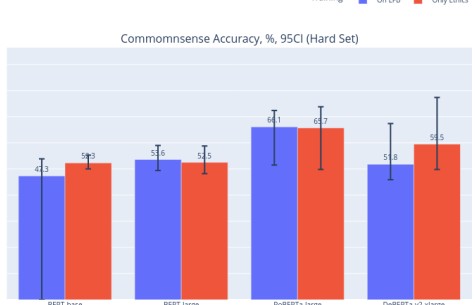

Figure 1: Accuracy values for the Commonsense split of the ETHICS dataset Hendrycks et al. [2020]. See Table 1 for a tabular depiction of the data.

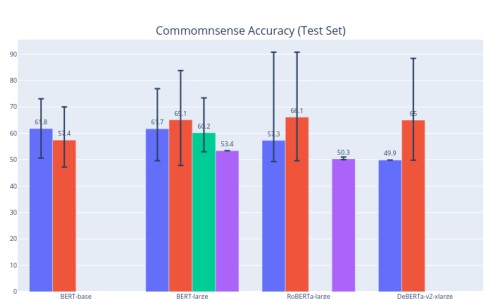
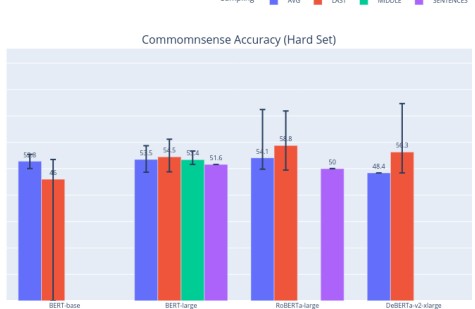

Figure 2: Graphical depiction of the different sampling methods' effect on accuracy on the Commonsense split of ETHICS Hendrycks et al. [2020].

| Model | Sampling | Fine-tuning | Brain Score Mean | Brain Score St. Dev. |
|---|---|---|---|---|
| BERT-large-cased | LAST | No fine-tuning | 0.217 | 0.096 |
| BERT-large-cased | LAST | ETHICS and fMRI | 0.213 | 0.095 |
| RoBERTa-large | LAST | No fine-tuning | 0.173 | 0.117 |
| RoBERTa-large | LAST | ETHICS | 0.145 | 0.097 |
| RoBERTa-large | LAST | ETHICS and fMRI | 0.156 | 0.112 |
| RoBERTa-large | LAST | ETHICS then fMRI | 0.144 | 0.113 |
| DeBERTa-v2-xlarge | LAST | No fine-tuning | 0.271 | 0.094 |
| DeBERTa-v2-xlarge | LAST | ETHICS | 0.266 | 0.095 |
| DeBERTa-v2-xlarge | LAST | ETHICS and fMRI | 0.273 | 0.096 |
| DeBERTa-v2-xlarge | LAST | ETHICS then fMRI | 0.264 | 0.097 |
| DeBERTa-v2-xlarge | LAST | fMRI then ETHICS | 0.237 | 0.097 |

Table 2: Brain scores across models and different fine-tuning methods. We were unable to significantly increase brain-model correlation using any of the fine-tuning methods.

in specific domains. Secondly, we make our code available.[3] We believe that further work along the neuroconnectionist research agenda Doerig et al. [2023] will be useful generally, and hope that the preliminary evidence we provide here will help update others' research models on the ability to increase alignment in specific domains.

## Acknowledgments and Disclosure of Funding

The authors were originally brought together and worked on this project through the 8th cohort of the AI Safety Camp. Bogdan-Ionut Cirstea was funded by the Center for Long-term Risk (CLR). Many thanks to Austin Brockmeier for helpful discussions.

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

## A   Brain Scores and Cortical Maps

In Figure 3, we plot further brain scores across different layers. Table 2 shares the brain score values averaged across the entire model. Below, in Figures 4 through 12, we plot the cortical maps of different fine-tuning methods on different models, as well as the cortical map of the activations provided by NeuroSynth Yarkoni et al. [2011] for our four areas of interest: language, moral reasoning, theory of mind, and vision.

We used vision ROIs as a control group. Note that our models were not able to achieve better brain scores than the control group, meaniing that our experiment did not achieve the desired effect. Nevertheless, we believe that releasing the results of our experiments may help to inform future research about the necessity of larger datasets and more effective fine-tuning.

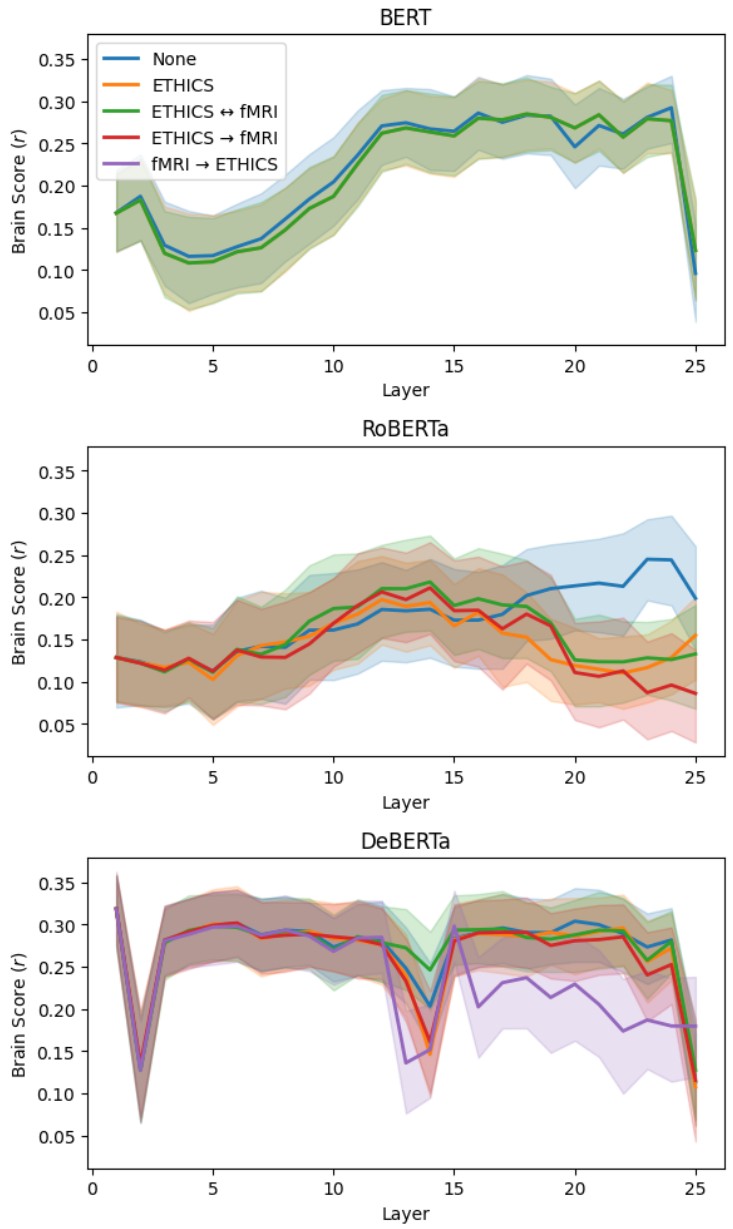

Figure 3: Brain scores across the hidden layers from bert-large-cased, roberta-large, and deberta-v2-xlarge across our different fine-tuning protocols.

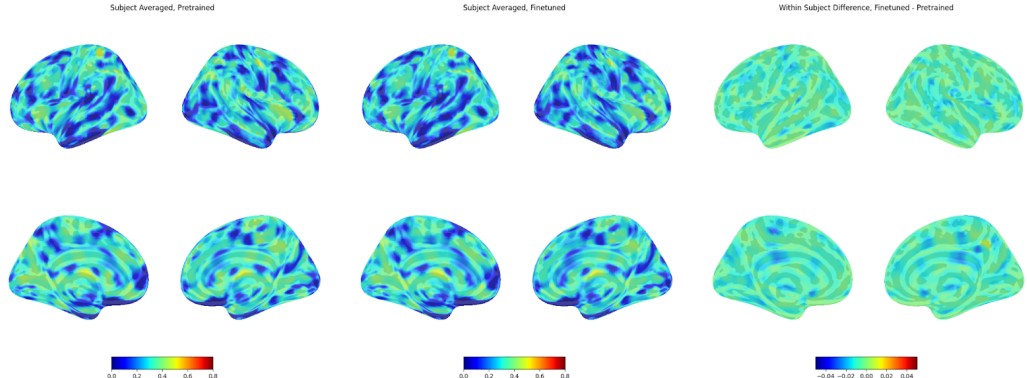

Figure 4: Subject and layer averaged CoD taken from bert-large-cased A) without fine-tuning on ETHICS or the fMRI recordings, B) with fine-tuning on ETHICS and fMRI recordings, and C) their difference.

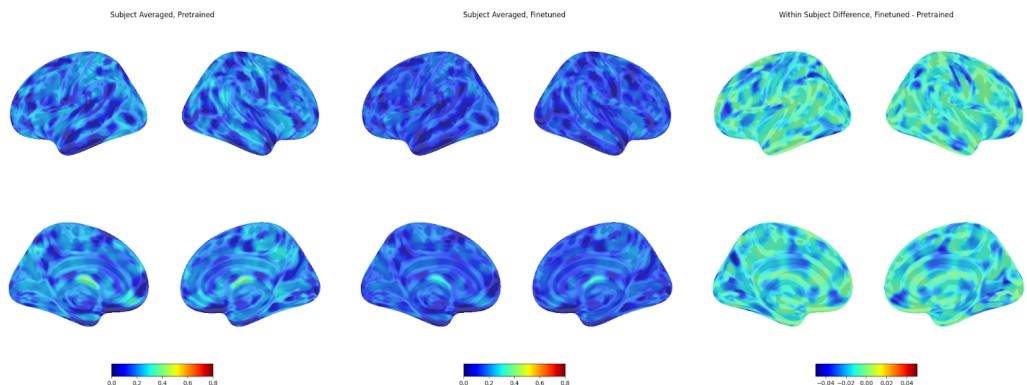

Figure 5: Subject and layer averaged CoD taken from roberta-large A) without fine-tuning on ETHICS or the fMRI recordings, B) with fine-tuning on ETHICS but not the fMRI recordings, and C) their difference.

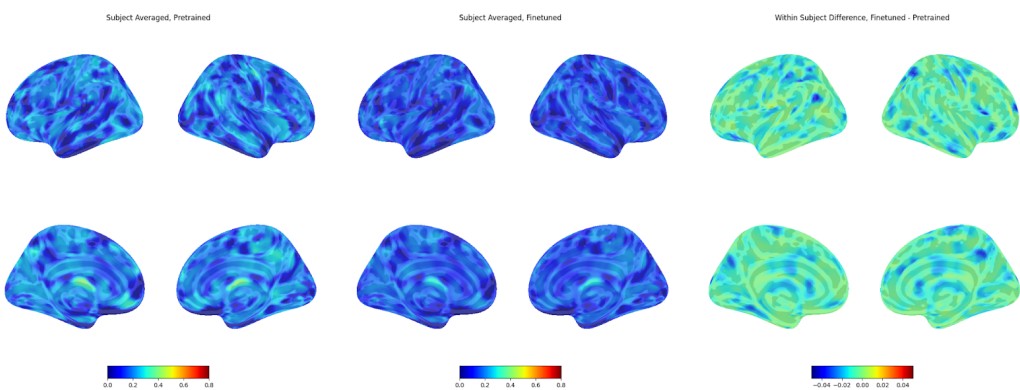

Figure 6: Subject and layer averaged CoD taken from roberta-large A) without fine-tuning on ETHICS and the fMRI recordings, B) with fine-tuning on both ETHICS and the fMRI recordings repeatedly, and C) their difference.

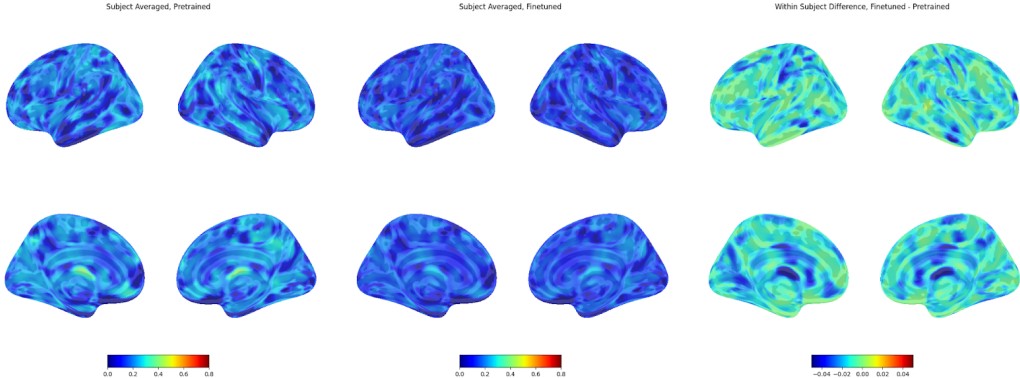

Figure 7: Subject and layer averaged CoD taken from roberta-large A) without fine-tuning on ETHICS and the fMRI recordings, B) with fine-tuning sequentially on ETHICS then on the fMRI recordings, and C) their difference.

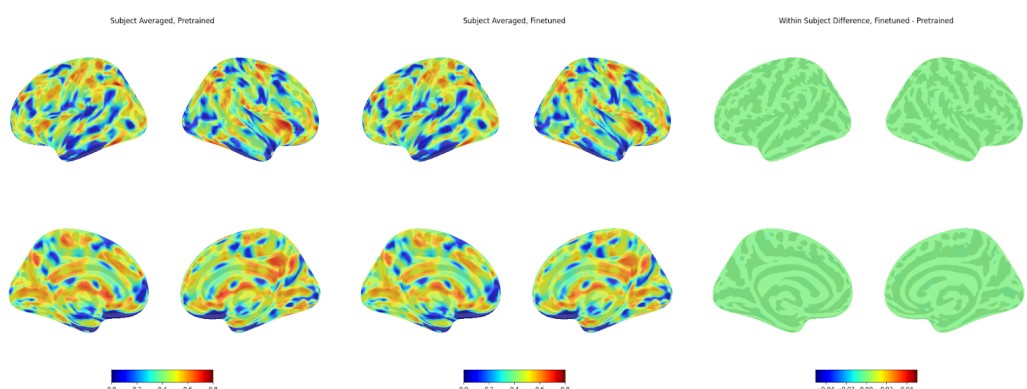

Figure 8: Subject and layer averaged CoD taken from deberta-v2-xlarge A) without fine-tuning on ETHICS and the fMRI recordings, B) with fine-tuning on ETHICS but not the fMRI recordings, and C) their difference.

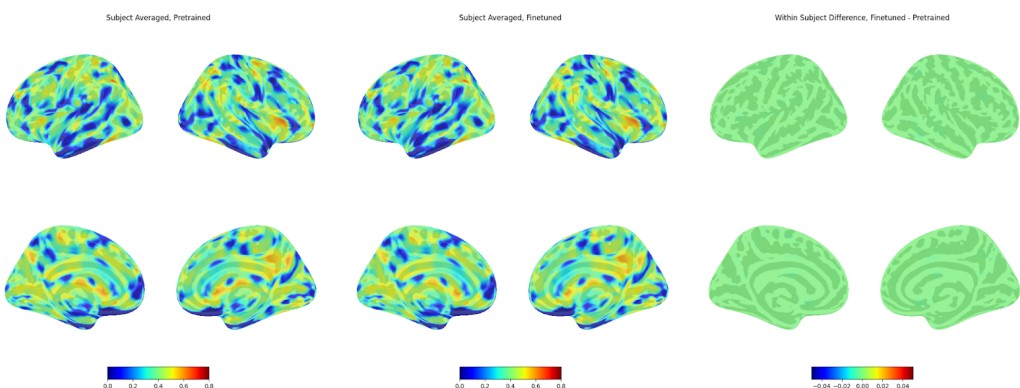

Figure 9: Subject and layer averaged CoD taken from deberta-v2-xlarge A) without fine-tuning on ETHICS and the fMRI recordings, B) with fine-tuning on both ETHICS and the fMRI recordings repeatedly, and C) their difference.

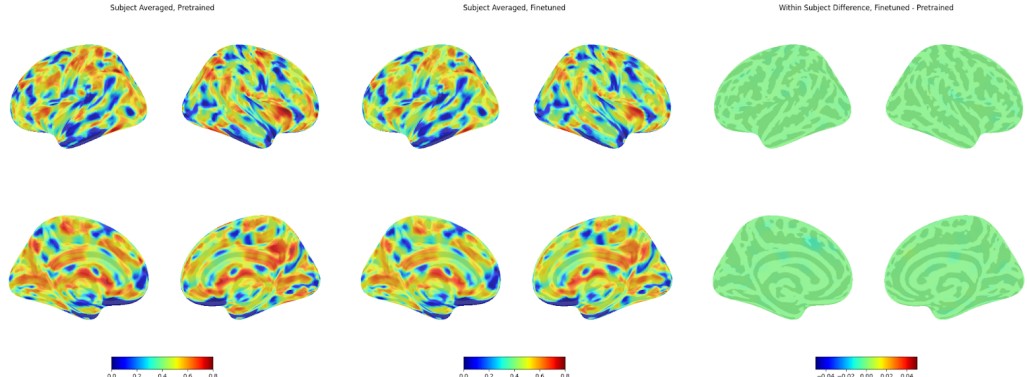

Figure 10: Subject and layer averaged CoD taken from deberta-v2-xlarge A) without fine-tuning on ETHICS and the fMRI recordings, B) with fine-tuning sequentially on ETHICS then on the fMRI recordings, and C) their difference.

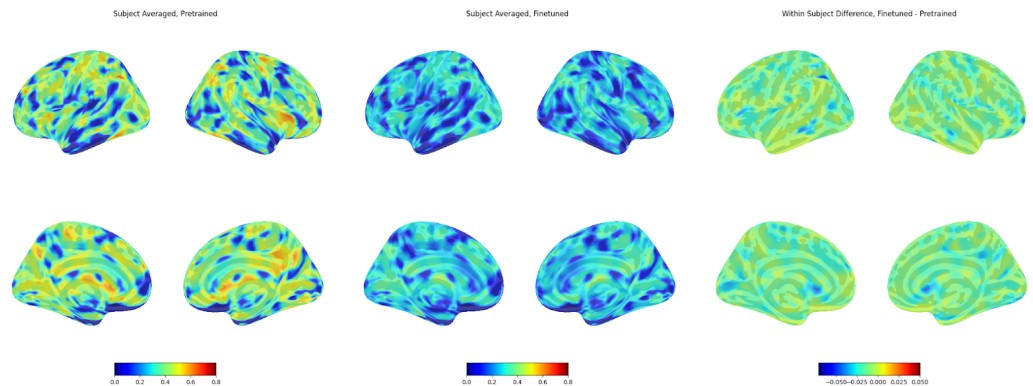

Figure 11: Subject and layer averaged CoD taken from deberta-v2-xlarge A) without fine-tuning on ETHICS and the fMRI recordings, B) with fine-tuning sequentially on fMRI recordings then on ETHICS, and C) their difference.

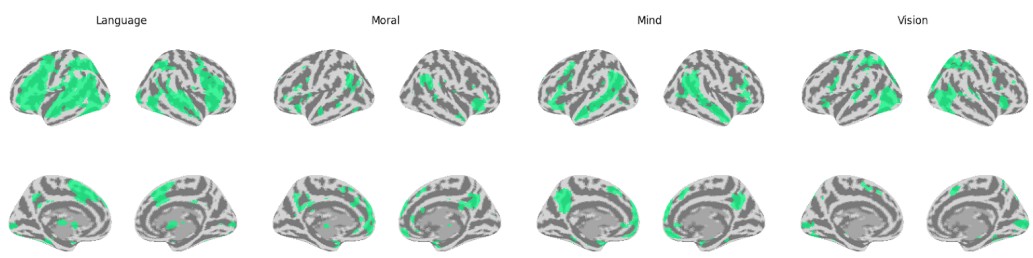

Figure 12: Functional activations taken from NeuroSynth meta-analyses for the terms language, moral, theory of mind, and vision, respectively.

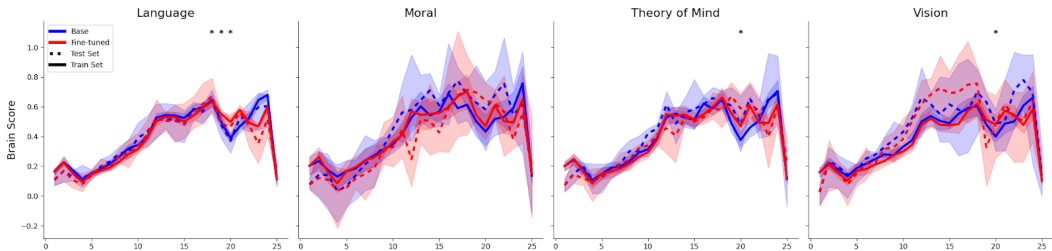

Figure 13: Scores of bert-large-cased across the term-based functional activations from NeuroSynth. The fine-tuning occurred over both ETHICS and the fMRI recordings, repeatedly. Asterisks indicate one-tailed Bonferroni corrected significance between the training pre-trained and fine-tuned scores if the fine-tuned scores are greater than the base scores.

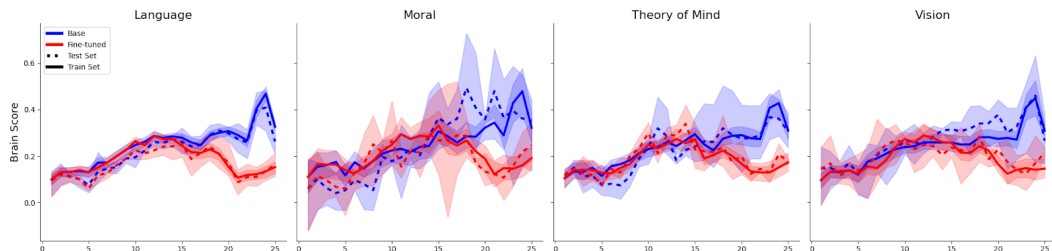

Figure 14: Scores of roberta-large across the term-based functional activations from NeuroSynth. The fine-tuning occurred over ETHICS but not the fMRI recordings. Asterisks indicate one-tailed Bonferroni corrected significance between the training pre-trained and fine-tuned scores if the fine-tuned scores are greater than the base scores.

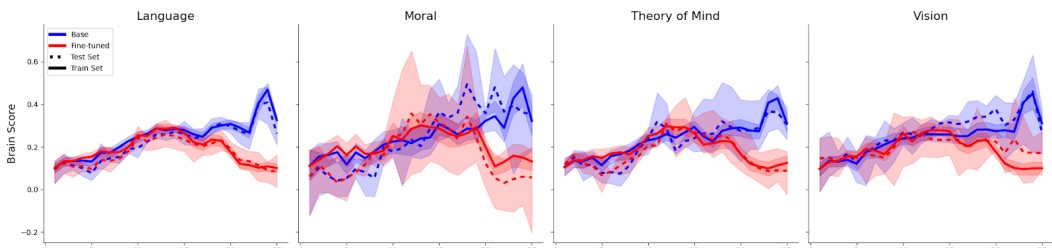

Figure 15: Scores of roberta-large across the term-based functional activations from NeuroSynth. The fine-tuning occurred over both ETHICS and the fMRI recordings, sequentially in that order. Asterisks indicate one-tailed Bonferroni corrected significance between the training pre-trained and fine-tuned scores if the fine-tuned scores are greater than the base scores.

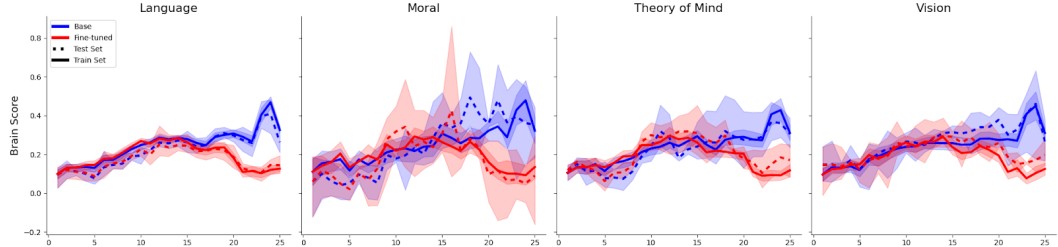

Figure 16: Scores of roberta-large across the term-based functional activations from NeuroSynth. The fine-tuning occurred over both ETHICS and the fMRI recordings, repeatedly. Asterisks indicate one-tailed Bonferroni corrected significance between the training pre-trained and fine-tuned scores if the fine-tuned scores are greater than the base scores.

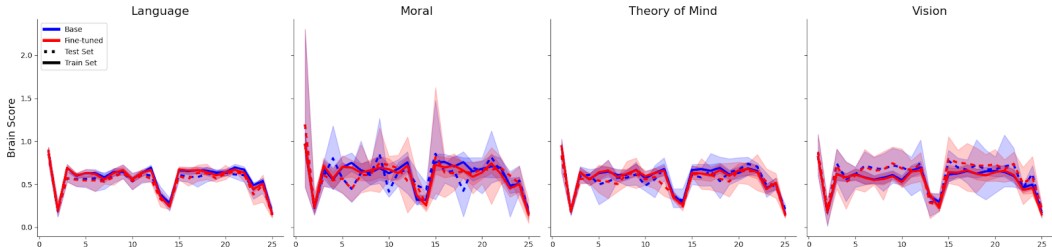

Figure 17: Scores of deberta-v2-xlarge across the term-based functional activations from NeuroSynth. The fine-tuning occurred over ETHICS but not the fMRI recordings. Asterisks indicate one-tailed Bonferroni corrected significance between the training pre-trained and fine-tuned scores if the fine-tuned scores are greater than the base scores.

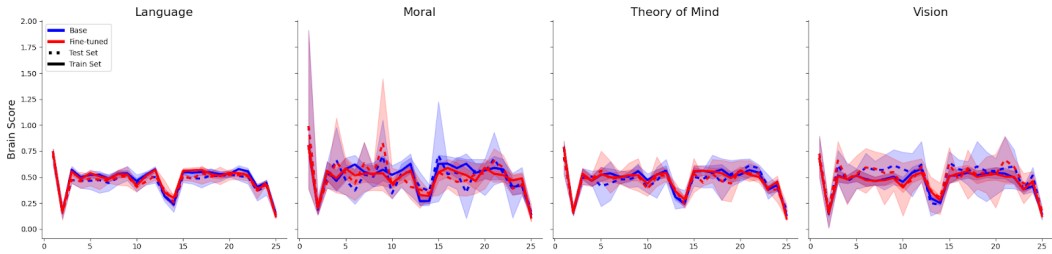

Figure 18: Scores of deberta-v2-xlarge across the term-based functional activations from NeuroSynth. The fine-tuning occurred over both ETHICS and the fMRI recordings, repeatedly. Asterisks indicate one-tailed Bonferroni corrected significance between the training pre-trained and fine-tuned scores if the fine-tuned scores are greater than the base scores.

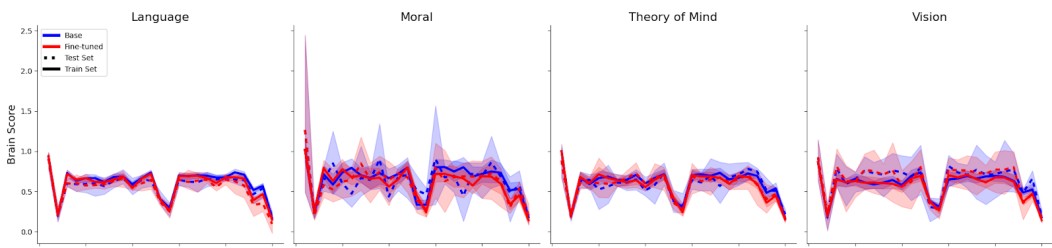

Figure 19: Scores of deberta-v2-xlarge across the term-based functional activations from NeuroSynth. The fine-tuning occurred over both ETHICS and the fMRI recordings, sequentially in that order. Asterisks indicate one-tailed Bonferroni corrected significance between the training pre-trained and fine-tuned scores if the fine-tuned scores are greater than the base scores.

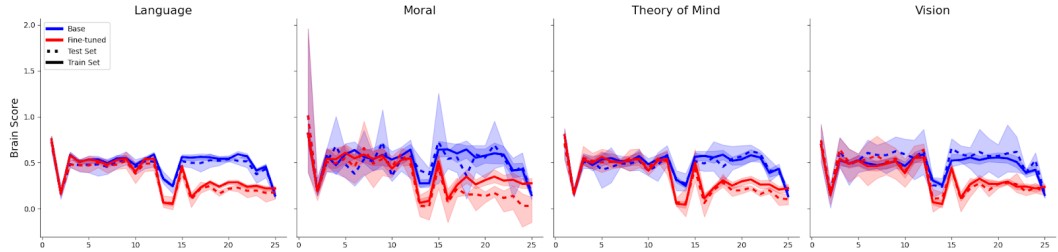

Figure 20: Scores of deberta-v2-xlarge across the term-based functional activations from NeuroSynth. The fine-tuning occurred over both fMRI recordings and ETHICS, sequentially in that order. Asterisks indicate one-tailed Bonferroni corrected significance between the training pre-trained and fine-tuned scores if the fine-tuned scores are greater than the base scores.

| Model | Sampling | Runs | CS Hard Set, % | | CS Test Set, % | |
|---|---|---|---|---|---|---|
| | | | mean ± STD | max | mean ± STD | max |
| BERT-base | AVG | 2 | **52.8 ± 3.9** | 55.5 | **61.8 ± 16.7** | 73.7 |
| BERT-base | LAST | 28 | 46.0 ± 16.3 | 53.6 | 57.4 ± 7.2 | 70.0 |
| BERT-large | AVG | 16 | 53.5 ± 3.0 | 59.6 | 61.7 ± 10.1 | 77.7 |
| BERT-large | LAST | 7 | **54.5 ± 4.8** | 61.8 | **65.1 ± 14.8** | 85.4 |
| BERT-large | MIDDLE | 3 | 53.4 ± 3.1 | 57.0 | 60.2 ± 12.4 | 74.5 |
| BERT-large | SENTENCES | 1 | 51.6 | 51.6 | 53.4 | 53.4 |
| RoBERTa-large | AVG | 11 | 54.1 ± 9.0 | 72.5 | 57.3 ± 16.4 | 90.9 |
| RoBERTa-large | LAST | 7 | **58.8 ± 11.2** | 72.2 | **66.1 ± 20.3** | 91.4 |
| RoBERTa-large | SENTENCES | 3 | 50.0 ± 0.1 | 50.1 | 50.3 ± 0.6 | 51.0 |
| DeBERTa-v2-xlarge | AVG | 1 | 48.4 | 48.4 | 49.9 | 49.9 |
| DeBERTa-v2-xlarge | LAST | 4 | **56.3 ± 13.6** | 76.6 | **65.0 ± 19.1** | 89.9 |

Table 3: Comparison of different sampling methods for fine-tuning on the fMRI dataset. Bolded values are the best accuracies per model.

## B  Additional Experiment Details

In Table 3 we provide a tabular depiction of the effect that different sampling methods had on our fine-tuning experiments and accuracy on the Commonsense split of the ETHICS dataset Hendrycks et al. [2020]. See Figure 2 for a graphical depiction.

