# OpenReview forum: "Inducing Human-like Biases in Moral Reasoning Language Models"
_NeurIPS.cc/2024/Workshop/UniReps — UniReps_

### Official Review · Reviewer_mmjp · 2024-09-28
**Review of Inducing Human-like Biases in Moral Reasoning Language Models**

**Rating:** 5
**Confidence:** 1

**Review:**

Quality:
The paper focuses on alignment of large language models fine-tuned for moral reasoning on behavioral data and/or brain data of humans performing the same task. It conducts solid experiments, and gives analysis of the use of fMRI data to fine-tune large language models (LLMs) for moral reasoning.

Clarity:
The paper is generally well-structured, with clear sections and explanations. The authors provide justifications for their answers to ethical considerations, ensuring transparency in their research process. Figures help readers to compare the results of experiments easily.

Originality:
The study's approach of aligning LLMs with human brain data using fMRI is original and contributes to the emerging field of ethical AI and natural language understanding.

Significance:
The significance of this research is less pronounced since authors are unable to significantly increase brain alignment on moral reasoning through fine-tuning methods. While they suggest their results could benefit future downstream work, this claim is unsupported. Additionally, their assertion that the study highlights the need for more data on moral reasoning and niche tasks feels somewhat self-evident and offers little in the way of new insights.

Pros:

Comprehensive and diverse experiments into brain alignment on moral reasoning.

Undertake the first measurement (BrainScore) of the similarity of the internal representations of biological brains (measured through fMRI) and large language models (LLMs).

Cons:

Experiment result does not bring consistent improve of accuracy on ETHICS.

Need more analysis into how the insights gained from the experiments could be applied to future study or real world applications.
Some typos, for example line 101 misses "Table" in "Table 2"

---

### Official Review · Reviewer_SfnA · 2024-10-06
**Paper proves that current fMRI data is not useful for aligning LLM models in the moral domain.**

**Rating:** 4
**Confidence:** 2

**Review:**

The reviewer is not familiar with neuroscience literature and thus trusts that the authors are using the best data available for such a task. Given the data, the setting seems reasonable, and the experiments are thorough.

The expectation that any representation of neural data would be functionally useful to steer LLM representations in a desired direction seems a little absurd. While the authors successfully show that such beliefs are unfounded, having such an expectation is questionable to begin with.

---

### Official Review · Reviewer_rAvw · 2024-10-06
**Offical Review**

**Rating:** 6
**Confidence:** 2

**Review:**

**Summary**

This paper investigates how fine-tuning large language models (LLMs) for moral reasoning tasks can improve their alignment with human brain activity (BrainScore).

**Strengths**

This study is the first to attempt aligning LLMs with human brain activity during moral reasoning by using fMRI data. This cross-disciplinary research offers a new approach for addressing the AI alignment problem, particularly in complex cognitive tasks.

**Weaknesses**

Although the models showed improved performance on the ETHICS benchmark, BrainScore did not see significant gains. Whether fine-tuning with only fMRI data or combining ETHICS and fMRI data, there was little improvement in the alignment between the model and human brain activity.

---

### Official Review · Reviewer_1uPA · 2024-10-07
**Review for: Inducing Human-like Biases in Moral Reasoning Language Models**

**Rating:** 6
**Confidence:** 2

**Review:**

This paper studies the alignment of BERT representations on moral reasoning tasks with corresponding fMRI data recording the brain activations of humans passing moral judgements. The results seem to indicate that for the task of moral reasoning, aligning the representations of the classifier to brain activity does not help to improve the performance on the moral reasoning dataset.

The paper is well written and I agree that the research is interesting and worthwhile pursuing. To make it accessible to a wider audience, it would probably help to combine Sections 1 and 2 and to better motivate and explain some of the claims, e.g. how aligning representations for moral reasoning tasks would be relevant for ToM or why it would be relevant for AI alignment.

I also think the finding that "improving brain-model alignment on moral reasoning by fine-tuning on relevant fMRI data does not consistently improve accuracy on [the textual moral judgement dataset]" might be due to the fact that according to the measured BrainScores none of the training/fine-tuning actually improved the brain-model alignment, according to Table 2 and Figure 3.

Being outside the field, I cannot pass judgement whether the observed "unlearnability" and/or failure to align the representation are inherently due to the difference in tasks/modalities/representations or due to the chosen method of representing the fMRI data in such a way that it is learnable by BERT.

Nonetheless I think this work in progress deserves to be presented as an extended abstract if there is space to do so in this workshop.

---

### Decision · Program_Chairs · 2024-10-10

**Decision:**

Accept

**Comment:**

In light of the reviewers' feedback and relevancy of the submission, we are pleased to accept this paper for presentation at UniReps 2024. We kindly ask the authors to incorporate the reviewers' suggestions and feedback in the final camera-ready version of the manuscript.